# Laser-Processed PEN with Au Nanowires Array: A Biocompatibility Assessment

**DOI:** 10.3390/ijms231810953

**Published:** 2022-09-19

**Authors:** Jana Pryjmaková, Barbora Vokatá, Petr Slepička, Jakub Siegel

**Affiliations:** 1Department of Solid-State Engineering, University of Chemistry and Technology Prague, 166 28 Prague, Czech Republic; 2Department of Microbiology, University of Chemistry and Technology Prague, 166 28 Prague, Czech Republic

**Keywords:** laser-treatment, periodic structures, gold nanowires, nanocomposites, biocompatibility

## Abstract

Although many noble metals are known for their antibacterial properties against the most common pathogens, such as *Escherichia coli* and *Staphylococcus epidermidis*, their effect on healthy cells can be toxic. For this reason, the choice of metals that preserve the antibacterial effect while being biocompatible with health cells is very important. This work aims to validate the effect of gold on the biocompatibility of Au/Ag nanowires, as assessed in our previous study. Polyethylene naphthalate (PEN) was treated with a KrF excimer laser to provide specific laser-induced periodic structures. Then, Au was deposited onto the modified PEN via a vacuum evaporation method. Atomic force microscopy and scanning electron microscopy revealed the dependence of the surface morphology on the incidence angle of the laser beam. A resazurin assay cytotoxicity test confirmed safety against healthy human cells and even cell proliferation was observed after 72 h of incubation. We have obtained satisfactory results, demonstrating that monometallic Au nanowires can be applied in biomedical applications and provide the biocompatibility of bimetallic Au/AgNWs.

## 1. Introduction

Targeted applications specify the chemical composition of the nanomaterial necessary to obtain the required properties and functions. Structured metals represent an attractive group of materials that are applicable in the biomedical field due to their unique chemical structure, which influences their properties on the nanoscale. Gold and its nanoforms have been known since the beginning of the modern age. In the sixteenth century, a solution of gold called *Aurum Potabile* (which was administered orally) was used to treat epilepsy. Even with the gradual development of medicine, the use of gold as a drug has not been abandoned. It was used to treat the most common disease of the nineteenth century, syphilis. In the 1920s, gold found therapeutic use once more in the treatment of tuberculosis, where it was used as a bacteriostatic agent against *Mycobacterium tuberculosis*, which is the cause of this disease [1]. Gold and its compounds are still used in the treatment of rheumatic diseases [2,3].

Today, the applicability of gold, especially gold nanomaterials, is much wider. Gold occupies an exceptional position in the field of nanomaterials since its optical and electrical properties depend on size and shape. When exposed to an electromagnetic wave, gold nanoparticles can exhibit localized surface plasmon resonances (LSPRs), caused by electron oscillations in the free conduction band [4]. The oscillation of the electrons subsequently leads to the formation of a net charge at the particle boundaries due to their delocalization upon radiation. Depending on the absorption or scattering of radiation, electromagnetic or optical phenomena may occur. This phenomenon can be applied in cancer thermal treatment [5,6], bioimaging [7], or biosensing [8]. Gold nanoparticles (AuNPs) are used for more accurate diagnosis, more effective targeted therapy, and molecular mapping. Their unique properties can be enhanced by the action of a laser, promoting the destruction of bacteria or even cancer cells [9,10]. In general, gold nanomaterials can serve as nanocarriers [11], nanosensors, nanotrackers [12], or even as part of a tissue scaffold to increase the ability of scaffold–tissue interaction for cardiac, neural, or bone cells [13].

Furthermore, AuNPs provide excellent biocompatibility and significantly low cytotoxicity [14], which are important factors for cell adhesion and proliferation. AuNPs in the form of hybrid scaffolds have been successfully used in bone formation to provide osteogenic proliferation. The study by Suarasan et al. [15] showed better proliferation and even the differentiation of osteoblasts with increased concentration of AuNPs/gelatin composite. Another study [16] introduced an innovative scaffold combined with AuNPs/polydopamine/polycaprolactone (PCL), capable of initiating osteogenesis. Gold nanomaterials also have an irreplaceable position in cardiac tissue engineering. Navaei et al. [17] prepared a hydrogel based on gelatine methacrylate enhanced with gold nanorods, which increased mechanical stiffness and promoted the electrical conductivity of the scaffold. Cell tests showed the excellent adhesion properties and viability of cardiomyocytes with metabolic activity. Similar properties were observed in the case of AuNPs/PCL/gelatine nanocomposite that was prepared by electrospinning and AuNPs evaporation [18], while AuNPs/PCL, with the addition of chitosan, provided neural Schwann cell proliferation and the required conductivity. In addition to fully synthetic scaffolds, a hybrid scaffold composed of biological tissue and AuNPs can also be prepared. An example of a hybrid type of scaffold can be cardiac tissue patches, made from decellularized omental tissue with deposited AuNPs [19]. These cardiac patches can be used to regenerate damaged heart tissues after myocardial infarction; gold provides electrical conductivity, and the omental tissue ensures the biocompatibility of the final scaffold.

In addition to pure gold nanomaterials, nanocomposites based on the combination of gold with other metals, e.g., silver (the most well-known antibacterial agent) also show remarkable results. As gold and silver nanomaterials have different properties, their mutual combination makes it possible to create a unique material with wider applicability, wherein silver has a function as an antibacterial agent, and gold plays a key role in tissue interactions. Many articles have been reported on the toxicity of nanosized silver [20,21,22], so the creation of a bimetallic Au-Ag nanomaterial can represent an effective solution to reduce silver toxicity while maintaining antibacterial activity. The study by Yang et al. [23] showed the great biocompatibility and significant antibacterial activity of core-shell Au@AgNPs. Synthesized Au@AgNPs manifested even higher antibacterial efficacy with lower Ag concentration, compared to monometallic AgNPs. Other examples of increasing biocompatibility are Au-AgNPs prepared from *Lansium domesticum* extract [24] or Au-Ag-alloy NPs, synthesized using isonicotinylhydrazide [25].

On the basis of the available literature and our previous study dealing with Au/Ag bimetallic nanowires [26] with the provided biocompatibility, we decided to examine the role of gold nanowires. For this reason, the aim of this work is to study the behavior of monometallic AuNWs in a biological environment; we assumed that the biocompatibility of Au/AgNWs after a long incubation time was caused by the presence of gold. We deposited Au on a laser-patterned polyethylene naphthalate (PEN) using the vacuum evaporation technique. After a thorough surface analysis proving that the AuNW structure is fairly comparable with that of bimetallic nanowires, the cytotoxicity tests were performed to determine the biocompatibility of Au/PEN samples.

## 2. Results and Discussion

### 2.1. Surface Characterization

The surface morphology of pristine polyethylene naphthalate (PEN), PEN 0/22.5/45°, and Au/PEN 0/22.5/45° was studied using atomic force microscopy (AFM). Figure 1 shows the AFM scans of the sample surfaces; Table 1 shows the derived surface parameters such as the average roughness of the surface (*R*_a_), periodicity (*Λ*), and height (*h*). It is obvious that the surface morphology of the laser-treated PEN significantly differs from that of the pristine PEN. The periodically arranged ripples (LIPSS) were formed as a consequence of the laser beam’s interaction with the polymer surface. This interaction is initiated by the action of pulses of linearly polarized UV radiation on the surface of the polymer substrate able to absorb it in this range (e.g., via the presence of an aromatic compound). The interference of the incidence of the laser beam and surface-scattered light is the main mechanism of the whole process (nanostructure formation). This interference caused an inhomogeneity in the distribution of energy density on the surface and started the thermal process to achieve a stable state with the formation of LIPSS [27,28,29,30,31]. One can see that the design and parameters of the created LIPSSs are distinguished in addiction on the incidence angle of the laser beam (0/22.5/45°). These results are in line with those obtained in the study by Slepicka et al. [32], who dealt with the angle-dependent nanostructuring of polyethylene terephthalate (PET). 

The increasing trend of recorded surface parameters with an applied incidence angle is clearly seen from the data presented in Table 1. Interestingly, there are also differences in the parameter values of the rippled PEN and rippled PEN with AuNWs. The surface morphology after the deposition of Au was preserved; however, the periodicity and the height decreased slightly. Although the characteristics changed, an increasing trend was still preserved. This phenomenon is known as a shadow effect [33] and occurs typically during nanowire deposition on a rippled polymer under specific experimental conditions [34,35,36]. 

To visualize the surface of the samples, scanning electron microscopy was used. Images of the gold-deposited samples are shown in Figure 2. It is obvious that after the Au deposition process, gold nanowire arrays formed on the PEN surface. One can see an increasing trend of periodicity in correlation with an increasing incidence angle, resulting from Equation (1) [37]:(1)$$\Lambda =\frac{\lambda }{n-\sin\theta }$$
where *λ* is the wavelength of laser light used, *n* represents the effective refractive index, and *θ* is the incidence angle of the laser beam. Since the BSE detector was used, Au nanowires are visualized as intense lines due to their different atomic weight; heavier elements are shown as bright areas on the sample surface. Thus, the AuNWs are situated on one side of the ripples, in accordance with the shadow effect. 

X-ray photoelectron spectroscopy (XPS) determined the elemental composition of PEN, PEN 0/22.5/45°, and Au/PEN 0/22.5/45°. The concentrations of gold (Au), oxygen (O), and carbon (C) are summarized in Figure 3A,B. After laser irradiation, no significant changes in elemental composition were observed. After gold deposition, 20–30 at. % Au was presented on Au/PEN 0/22.5/45°, which indicates the successful formation of AuNWs. Differences in the amount of gold could be attributed to the specific surface morphology and the experimental setup of gold deposition. The surface of PEN was pre-treated with the laser prior to the deposition of the gold to create gold nanowires. The samples were placed in the sample holder in the same way: the LIPSSs were perpendicular to the metal flow. However, in the case of PEN, with tilt angles of 22.5° and 45°, the underlying PEN patterns were not collinear, so it was unable to preserve their perpendicular orientation to metal flow.

Since wettability is an important biocompatibility parameter that affects the cell attachment process, the contact angle (CA) was measured for PEN, PEN 0/22.5/45°, and Au/PEN 0/22.5/45°. The CA, as a function of the incidence angle, is shown in Figure 4. As expected, the CA of the laser-treated PEN did not differ from those referred to in an earlier study [26,38]. In the case of laser-treated PEN at incidence angles of 22.5° and 45°, the CA reached a value of approximately 73°, which is fairly close to that of pristine PEN. The CA of AuNWs was only 83° for all modification angles, so the final metal-polymer composite exhibits a hydrophilic character, compared to Au/AgNWs and CuNWs [38]. This finding is quite interesting, because the CA of bimetallic Au/AgNWs was approximately 105°, while the CA of monometallic AuNWs and AgNWs [36] was below 90°. From the measured heights of the structures (derived via AFM measurement), it was even assumed that the surface would show a rather hydrophilic character, which could correspond to the Cassie–Baxter wetting state [39]. The explanation for this can be found in the distribution of the metal. In the case of bimetallic nanowires, the metals covered a larger area of ripples compared to that of the monometallic nanowires, which were located just on one side of the nanostructures, so the water droplets were in contact with the metallic and polymeric surface at the same time.

### 2.2. Cytotoxicity Tests

To determine the effect of Au on the biocompatibility of Au/AgNWs supported on PEN [26], the cytotoxicity of selected samples (PEN 22.5° and Au/PEN 22.5°) was tested. The results are shown in Figure 5. As expected, the control samples (standard tissue culture polystyrene) showed the highest values, which is consistent with the study by Pessková et al. [40]. After 24 h of incubation, 60% of the cells seeded on the surface of PEN 22.5° and Au/PEN 22.5° exhibited metabolic activity. This lower percentage (compared to the values after 48 and 72 h of incubation) can be explained by the gradual adaptation of the cells to the rough surface. Cell adaptation depends on surface topography, roughness, and stiffness [41]. Moreover, the crucial point relates to the types of cells for which the final material is designed. One can see that the number of viable cells reached almost 100% with an increasing period of incubation. After 72 h of incubation, Au/PEN 22.5° samples met the eligibility conditions for the adhesion and metabolic activity of cells.

At this point, we can state that gold supports cell adhesion and proliferation in its two forms—monometallic (Au) and bimetallic (Au/Ag, see [19]) nanowires. As disclosed by Polivkova et al. [21], monometallic silver nanowires exhibited a pronounced cytotoxicity effect on mouse embryonic fibroblasts. Their applications for tissue engineering or coating medical devices are limited by the period of contact with the tissue. The solution can be found in the combination of gold and silver to ensure the biocompatibility of the final material, offering a wide range of applications. 

## 3. Materials and Methods

### 3.1. Materials and Apparatus

A polyethylene naphthalate foil (PEN, thickness 50 µm, Goodfellow, Ltd., Huntington, UK) was irradiated with a KrF excimer laser (COMPexPro 50 F, Coherent, Inc., Santa Clara, CA, USA) according to the procedure described in our previous study [26]. Irradiation was performed at incidence angles of 0, 22.5, and 45°. In this way, samples with laser-induced periodical surface structures (LIPSSs), differentiated by surface characteristics, were prepared. 

The laser-treated PEN was coated with gold using the vacuum evaporation method. Gold evaporation in the form of pellets (3.18 × 3.18 mm^2^, purity 99.99%, Safina a.s., Vestec, Czech Republic) was carried out in a LEYBOLD-Heraeus vacuum evaporation apparatus (Univex 450, Cologne, Germany). The deposition of Au led to the formation of monometallic nanowires. The measurement of nanowire thickness was provided by the procedures referred to in [26], using the scratch method [42]. The arithmetical mean and standard deviation were calculated. The deviation did not exceed 5%.

### 3.2. Analytical Methods

The chosen surface analyses were carried out in accordance with the analyses given in a previous study dealing with bimetallic nanowires [26]; therefore, the instrumentation and experimental conditions are mentioned briefly. 

The surface morphology of the pristine polymer (PEN), rippled PEN (PEN 0/22.5/45°), and rippled PEN coated with Au nanowires (Au/PEN 0/22.5/45°) was studied with atomic force microscopy using the Dimension ICON (Bruker Corp., Billerica, MA, USA). Data and parameters such as average surface roughness (*R*_a_), height (*h*), and periodicity (*Λ*) were acquired using the NanoScope Analysis v1.4 software (Bruker, Billerica, MA, USA). Arithmetical means and standard deviations, which did not exceed 5%, were calculated. 

The visualization of rippled PEN with deposited AuNWs was performed using scanning electron microscopy (SEM LYRA3 GMU, Tescan, Brno, Czech Republic) at a voltage of 5 kV. The backscattered electron (BSE) signal was recorded.

To study the chemical composition of the samples, an X-ray photoelectron spectroscopy (XPS) was used. The samples were analyzed using an ESCAProbeP spectrometer (Omicron nanotechnology GmbH, Taunusstein, Germany) with a monochromatic X-ray beam source. To obtain representative XPS spectra, measurements were made at 90° with respect to sample orientation. Data analysis was performed using CasaXPS software. Data such as the concentration of elements (given in at.%) were summarized in a graph.

To determine the hydrophobic/hydrophilic character of the sample surfaces, the contact angle (CA) was measured using the sessile drop method using the KRÜSS DSA 100 goniometer (KRÜSS, Hamburg, Germany). Ten droplets of distilled water (software-controlled volume of 2 µL) were applied to the surface of the sample using the automatic pipette. Then, KRÜSS Advance software v2.0 calculated the Cas by the application of the three-point method. The final Cas were given as arithmetical means and standard deviations.

### 3.3. Cytotoxicity Tests

The toxicity of the rippled samples with AuNWs was tested in primary human lung fibroblast (MRC-5, American Tissue Culture Collection, Manassas, VA, USA). Human primary fibroblasts were incubated in Minimal Essential Medium Eagle (MEM, Sigma-Aldrich, St. Louis, MO, USA) supplemented with 2 mM L-glutamine (a stable dipeptide, sourced from Sigma-Aldrich, St. Louis, MO, USA) under standard conditions (37 °C, 5% CO_2_).

Samples (PEN 22.5° and Au/PEN 22.5°) with human health cells were prepared as described by the authors of [26]. In this case, cell viability was measured using a resazurin assay [43]. The medium was removed, then the samples were washed with PBS and incubated with a resazurin solution (final concentration 25 μg/mL) in a medium without phenol red (MEM, Gibco, Thermo Fisher Scientific, Waltham, MA, USA) and incubated for 4 h. Thereafter, fluorescence was measured using the Fluoroskan Ascent (Thermo Labsystems, Waltham, MA, USA) and excitation and emission wavelengths of 560 and 590 nm, respectively. Cells that were grown on standard tissue culture polystyrene (TCPS) were used as control. All samples were prepared in triplicate. Cell viability was represented as a percentage of metabolic activity of control cells. Mean values and standard deviations were calculated.

## 4. Conclusions

We chose an easy and effective approach by which to prepare self-organized monometallic nanowires using a laser surface modification and a vacuum evaporation technique. The results of the AFM analysis reported here confirm the formation of periodically arranged ripples with different surface characteristics, depending on the experimental setup (tilted laser-beam geometry). The SEM and XPS analysis showed isolated AuNWs that were located on one side of the ripples. After the deposition of gold, the contact angle reached 83°, indicating the creation of a surface with a hydrophilic character. These surface conditions were favorable for adhesion and cell proliferation. These findings might help to solve the problem of the cytotoxic effect of silver nanomaterials, which are among the most effective antibacterial agents. The presence of gold can decrease the toxicity of nano-sized silver on eukaryotic cells and preserve the antibacterial effect of the nanocomposite simultaneously in bimetallic Au/Ag. 

## Figures and Tables

**Figure 1 ijms-23-10953-f001:**
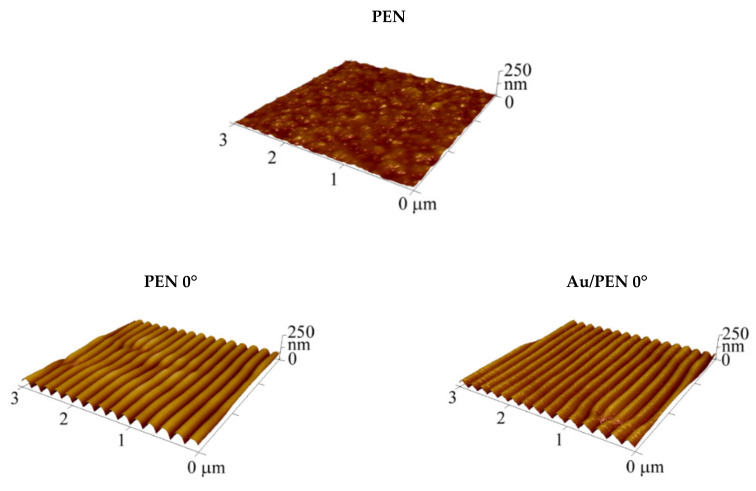
Three-dimensional atomic force microscopy images of pristine polyethylene naphthalate (PEN) and laser-treated PEN, shown at different angles before (PEN 0/22.5/45°) and after Au deposition (Au/ PEN 0/22.5/45°).

**Figure 2 ijms-23-10953-f002:**
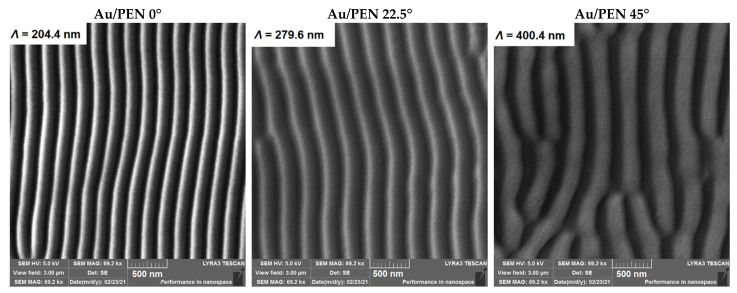
Images of Au/PEN 0/22.5/45° sample surfaces with appropriate periodicity, obtained by SEM.

**Figure 3 ijms-23-10953-f003:**
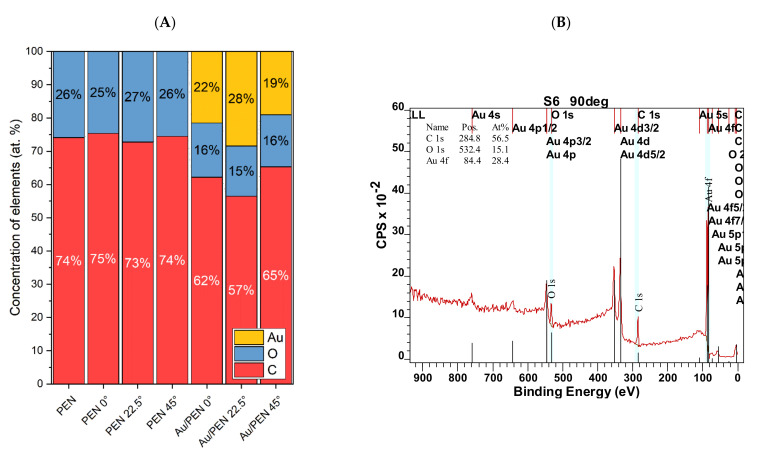
Concentrations of Au (gold), O (oxygen), and C (carbon) on the surface of the PEN (pristine), PEN 0/22.5/45°, and Au/PEN 0/22.5/45° samples expressed in at. % (**A**). Example of the XPS spectrum of Au/PEN 22.5° with atomic concentrations of elements (**B**).

**Figure 4 ijms-23-10953-f004:**
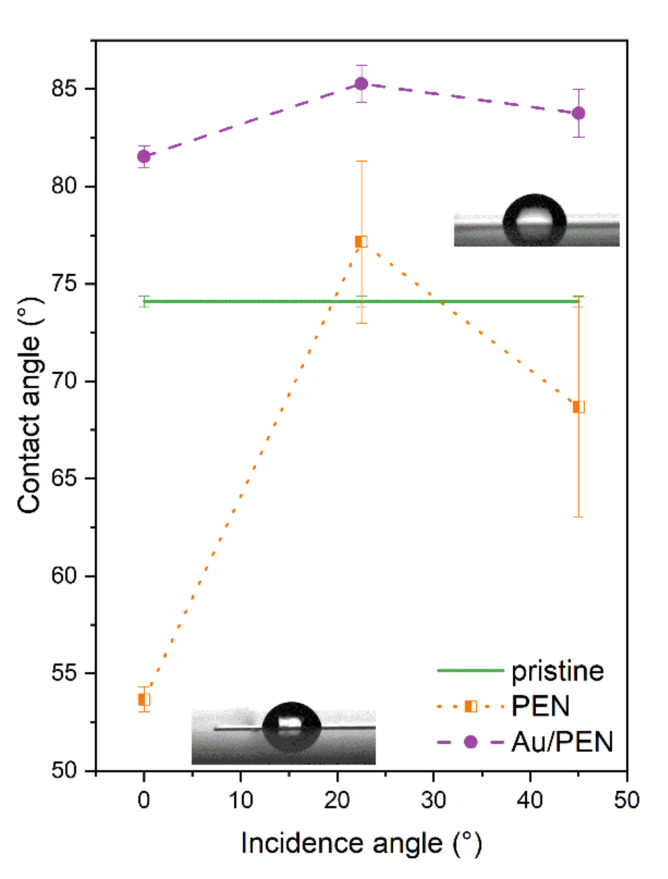
Dependence of contact angle on the incidence angle of laser light, preceding the Au metallization for the PEN (pristine), laser-treated PEN (PEN 0/22.5/45°), and laser-treated PEN with deposited AuNWs (Au/PEN 0/22.5/45°).

**Figure 5 ijms-23-10953-f005:**
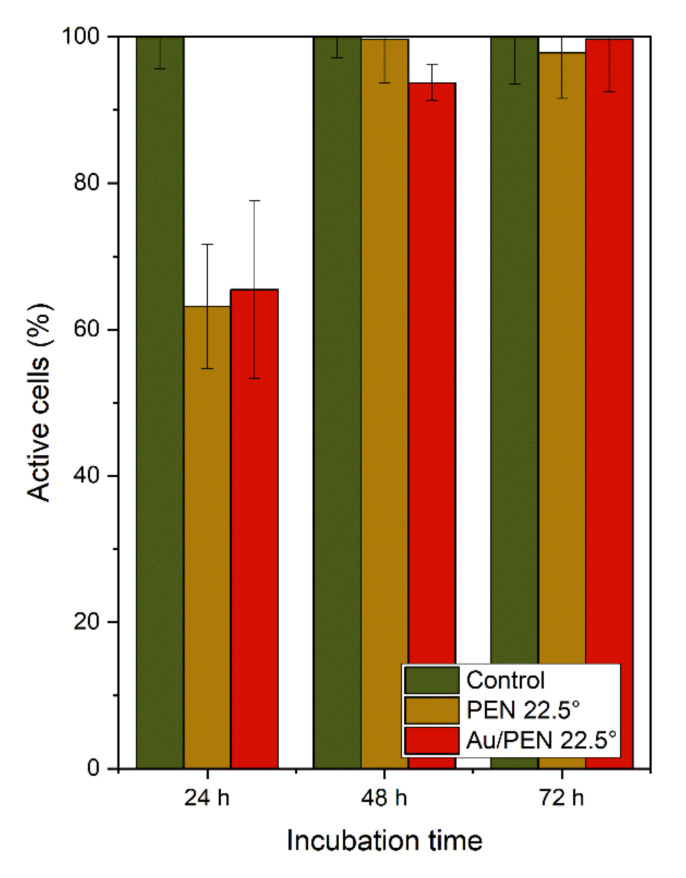
The decreasing cytotoxicity of PEN 22.5° and PEN 22.5° with AuNWs (Au/PEN 22.5°) after 24, 48, and 72 h of incubation. The cell viability was represented as a percentage of the metabolic activity of the control cells.

**Table 1 ijms-23-10953-t001:** Summary of the surface characteristics derived from AFM analysis: surface roughness (*R*_a_), periodicity (*Λ*), and height (*h*).

Sample	*R*_a_ (nm)	*Λ* (nm)	*h* (nm)
PEN	4.5	-	-
PEN 0°	16.0	207.6 ± 4.5	69.9 ± 2.2
PEN 22.5°	19.8	287.8 ± 3.5	88.1 ± 4.0
PEN 45°	34.3	339.6 ± 4.1	119.3 ± 2.7
Au/PEN 0°	14.9	204.4 ± 4.3	61.9 ± 3.2
Au/PEN 22.5°	20.6	279.6 ± 3.3	78.4 ± 2.9
Au/PEN 45°	40.6	400.4 ± 3.4	108.7 ± 2.7

## Data Availability

The data presented in this study are available on request from the corresponding author.

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
