# Peer review of "Laser-Processed PEN with Au Nanowires Array: A Biocompatibility Assessment"

_ijms, 2022, doi:10.3390/ijms231810953_

Round 1
Reviewer 1 Report
-- The laser treated PEN shows array structure, which is stated by the author by "interaction o f the laser beam with the polymer surface". I wonder what kind of "interaction" can generate this uniform structures in nanoscale? The interpretation should be added into the main text.
--Why the hight of Au/PEN/45 is increased rather than the decrease of the other sample?
--The details of measurement for contact angle should be mentioned, such as how to control drop volume as 2 microliter, how to determined the contact angle from raw image, how it look like the typical sessile drop on the sample captured by the camera (use the image in the figures)
-- More importantly, the static contact angle is measured, for which any values are possible, due to the hysteresis of the sample substrates, particularly samples with structures. So, the advancing and receding contact angle should be provided. Otherwise, the plot in Figure 4 are highly meaningless, especially for the purple plot of Au/Pen.
--The XPS total spectra and spectra with individual peaks from the measurements are highly needed to support the results in Figure 3.
-- The cell image on the sample under optical microscopy should be provided, rather then "black" data points on the plots in Figure 5.
-- Since Ag is found to have cytotoxicity, then the Au coatings is very important in the present work. Then the question is that, how to ensure the Ag strips are totally covered by the Au layer?
-- Finally, the introduction part should discuss what the question the authors ask, and what is the current progress on this topic, rather than most of the introduction part is about the advantage of the Au metal. The reference DIRECTLY relevant to the topic of the manuscript should be provided.
Author Response
Please, see the attachment.
Sincerely yours
Jana Pryjmaková

Reviewer 2 Report
The paper is well written and the method has been described well.
Statistical analysis method and error(s) are missed in the numerical data.
Most importantly, there several works have been done in this field for the last decade. If authors believed their method/study is novel, they need to add supporting information in the manuscript. It is necessary to discussed why this study has novelty/originality.
Author Response

(The authors gave the same response as above.)

Round 2
Reviewer 1 Report
My concerns are all addressed properly, so that I recommend acceptance of the current form.
Reviewer 2 Report
The revised manuscript can be accepted